# Evaluating Urban Flood Resilience within the Social-Economic-Natural Complex Ecosystem: A Case Study of Cities in the Yangtze River Delta

Shiyao Zhu [1,2], Haibo Feng [2] and Qiuhu Shao [1,*]

1   School of Transportation and Civil Engineering, Nantong University, Nantong 226001, China
2   Department of Wood Science, University of British Columbia, Vancouver, BC V6T 1Z4, Canada
*   Correspondence: shaoqiuhu@126.com

**Abstract:** With global climate change and rapid urbanization, it is critical to assess urban flood resilience (UFR) within the social-economic-natural complex ecosystem in dealing with urban flood disasters. This research proposes a conceptual framework based on the PSR-SENCE model for evaluating and exploring trends in urban flood resilience over time, using 27 cities in the Yangtze River Delta (YRD) of China as case studies. For the overall evaluation, a hybrid weighting method, VIKOR, and sensitivity analysis were used. During that time, UFR in the YRD region averaged a moderate level with an upward trend. This distinguishes between the resilience levels and fluctuation trends of provinces and cities. Jiangsu, Zhejiang, and Anhui provinces all displayed a trend of progressive development; however, Shanghai displayed a completely opposite pattern, mainly because of resilience in the state dimension. During that time, 81.41% of cities exhibited varying, upward trends in urban flood resistance, with few demonstrating inverse changes. Regional, provincial, and city-level implications are proposed for future UFR enhancement. The research contributes to a better understanding of the urban complex ecosystem under flood conditions and provides significant insights for policymakers, urban planners, and practitioners in the YRD region and other similar flood-prone urban areas.

**Keywords:** social-economic-natural complex ecosystem; urban flood resilience; pressure-state-response model; urban flood management

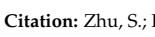



## 1. Introduction

Since the beginning of the 21st century, climate change has become a global concern, especially extreme climate events, which have caused various severe impacts on urban areas [1–3]. In China, climate extremes have increased, resulting in significant regional differences in precipitation changes and increased rainfall days [4,5]. The statistics show that, since 1951, the average precipitation in China has been increasing, especially from 2012 to 2021, which was the wettest decade in the past 70 years [6]. This kind of heavy rainfall process, especially extreme precipitation, lasts for a long time, accumulates a large amount of rainfall, and overlaps in the fall area, resulting in more frequent urban flood disasters [7,8]. In addition to climate change, human activities have also increased the probability of extremely heavy precipitation events, leading to urban floods and waterlogging disasters [9–11]. With the acceleration of urbanization, the speed of urban infrastructure construction cannot keep up with the pace of urban development [12]. Because of inadequate drainage regulations and flawed drainage systems [13,14], this kind of fast urbanization has encroached upon rivers and lakes, disrupted water systems, gradually eroded urban water retention areas, and caused recurrent urban flooding. Urban flood problems have become another major urban disease, following population crowding, traffic congestion, and environmental pollution.

In an effort to address these issues, the idea of resilience was developed, which offers fresh perspectives and creative solutions for managing and mitigating urban flooding [15,16]. Urban flood resilience (UFR) refers to the ability of urban systems to resist, recover, and sustain their normal functions when disrupted by flood disasters [17,18]. This concept has attracted research from governments and scholars in sociology, urban planning, disaster management, management, and other fields. Since 2015, the United Nations has developed a series of global policy processes and commitments to strengthen urban resilience, including flood resilience, such as *the 2030 Agenda for Sustainable Development* [19], the *Sendai Framework for Disaster Risk Reduction 2015–2030* [20], and *Making Cities Resilient 2030* [21]. Developing countries such as China have also realized the importance of improving urban resilience against major disasters, and China mentioned it for the first time in the *Proposal of Formulating the Fourteenth Five-Year Plan for National Economic and Social Development and the 2030 Long-Term Goals* [17].

Previously, UFR-related studies, including flood risk management, flood risk assessment, and flood mitigation strategies, were built on a solid foundation for the analysis of UFR [22–24]. Various frameworks to quantify UFR have been developed, either based on indicator systems [25,26], flood scenario simulations [27,28], or qualitative investigations [29,30]. Many studies consider UFR under the concept of urban resilience or take flooding as a disaster example to assess urban resilience [31–33]. Such comprehensive assessment index metrics for urban resilience usually include social, economic, infrastructure, community, and environmental dimensions [34,35], which can be used to assess UFR but are not always specific or relevant. Others consider UFR from the disaster process perspective, with an emphasis on particular flood events or stages, such as the resistance ability before the flood, the coping capacity during the flood, or the recovery capacity after the flood [36,37]. These studies are more simulation-based and time-sensitive and demand great data accuracy. However, flood disasters are a dynamic process; evaluating UFR from a single or multiple flood events is difficult and complex, and some of the indexes are difficult to quantify [7]. Following the assessment, the implications for policymaking, acting, and resilience improvement are not always clear. This requires holistic thinking about abilities to reflect the dynamic course of urban ecosystems against flooding.

In order to avoid one-sidedness in the assessment of UFR, it is necessary to establish a framework with considerations of the preparation, resistance, and recovery ability of the whole urban ecosystem under the flood cycle. Furthermore, a relative assessment of the urban ecosystem of a large area sharing similar basin characteristics and specific regional features can also help enrich the commentary points of resilient cities, put forward targeted policies, and promote regional sustainable development. Thus, this paper aims to establish a UFR evaluation framework considering the social–economic–natural complex. ecosystem as well as the flood cycle for a large region in China during a certain period. The results can provide guidance and suggestions to policymakers so they can formulate more targeted and practical plans for regional, provincial, and city-level urban flood management and resilience improvement.

## 2. Materials and Methods

### 2.1. Study Area

The Yangtze River, China's mother river, has been threatened by flood disasters for thousands of years. The Yangtze River Delta (YRD) is located at the lower reaches of the Yangtze River and is an alluvial plain formed by the siltation of sediments brought by the Yangtze River [38]. Since the 1980s, flood disasters have intensified, and it is the center of frequent water disasters in China. The YRD urban agglomeration is a highly developed economic zone in the eastern coastal area of China, with the highest population density and urbanization speed [8]. Because of its location in the plain and low-lying areas, the YRD region is sensitive to its climate and is separated into different air masses by the boundary line between subtropical and temperate climates. About 70% of the rainfall is concentrated in spring and summer, which makes these two seasons the most prone to

flood disasters [39]. During the flood season of 2016, the middle and lower reaches of the Yangtze River suffered the worst flooding since 1999, involving Hunan, Hubei, Anhui, Jiangxi, and Jiangsu provinces [40]. Considering its economic background and flood history, 27 cities from 3 provinces and 1 municipality in YRD were chosen to conduct an empirical analysis, as shown in Figure 1. The research period covers five years, from 2015 to 2019, which is consistent with the starting year of the "Development Plan for the Yangtze River Delta Urban Agglomeration". To avoid the impact of COVID-19, the years 2020 to 2022 were not considered in the model.

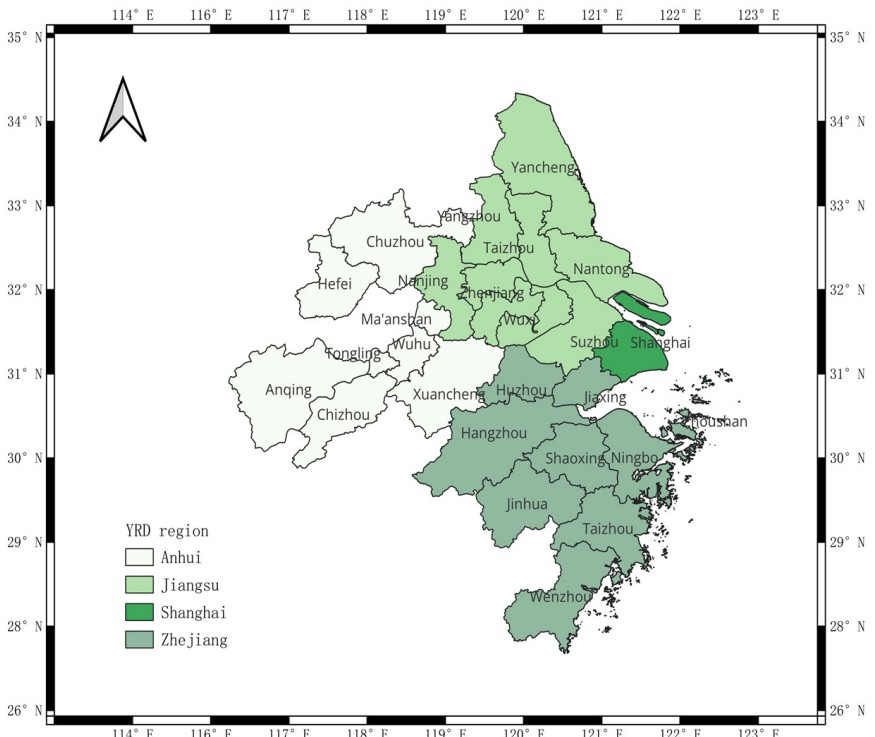

**Figure 1.** Provinces and cities in the YRD region.

### 2.2. Framework Establishment

Considering the complex ecosystem of cities, the flood disaster cycle, and the concept of urban resilience, a conceptual framework was built first to identify indicators for UFR evaluation. The pressure–state–response (PSR) model was used to determine the urban situation during flood disaster cycles [41], and the social–economic–natural complex ecosystem (SENCE) was adopted to show the complex ecosystem of cities. Both of these can systematically describe and analyze the interaction between society and the environment [42–44]. Because of global climate change and the frequent occurrence of various natural disasters and social events, some scholars have already begun to explore the applicability of the PSR model in disaster management, such as exploring the factors affecting the eco-environment during earthquakes and flooding [17,45] or assessing the risk for coal flood water inrush [46].

In general, the PSR-SENCE framework considers the risks of urban floods (pressure) on the environment; how they affect the quality and quantity of natural, economic, and social resources (state); and how society responds to these changes through natural, economic, and policy changes, as well as changes in awareness and behavior (response). Thus, three main dimensions with seven subdimensions are considered in this framework to evaluate UFR, including natural pressure, natural state, social state, economic state, natural response, social response, and economic response, as shown in Figure 2.

Urban flood resilience

| PSR | Pressure | State | Response |
|------|----------|-------|----------|
| SENCE | natural pressure<br>*Urban Flood* | natural state | natural response |
| | | economic state | economic response |
| | | social state | social response |

**Figure 2.** Conceptual framework.

### 2.3. Evaluation Indicators Identification

Based on the framework, a comprehensive systematic review (SR) was conducted to identify the primary indicators of UFR using related previous studies. Following a literature screening process based on SR, four major steps were adopted, including identification, screening, eligibility, and inclusion [47]. Indicators with a frequency of over 5 were considered primary indicators, and they were grouped according to the above framework with the Delphi method. From June to July 2022, taking into account the impact of COVID-19, 15 experts were invited to justify the indicators through video meetings, with detailed profiles in Table 1. Questionnaires and interviews were provided for the experts to judge and score the chosen indicators, and those indicators scoring in the top 80% were selected for the evaluation [48]. Eventually, 24 indicators under the framework were accepted as shown in Figure 3.

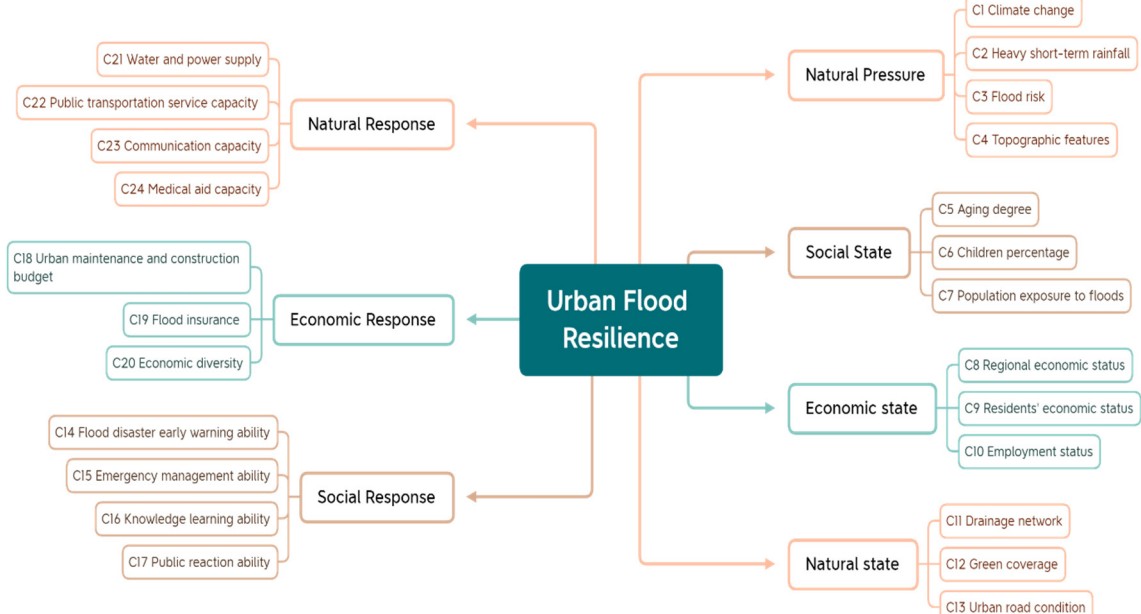

**Figure 3.** The indicators for urban flood resilience evaluation.

**Table 1.** Profiles of the experts.

| Basic Characteristics | | Percentage | Basic Characteristics | | Percentage |
|---|---|---|---|---|---|
| Gender | Male | 60% | Work experience | Over 10 years | 13.33% |
| | Female | 40% | | 7–9 years | 46.67% |
| Occupation | Government officer | 60% | | 5–6 years | 40% |
| | University professor | 26.67% | Education | Doctoral degree | 33.33% |
| | Related municipal company manager | 13.33% | | Master's degree | 40% |
| | | | | Other | 26.67% |

### 2.3.1. Indicators in the Pressure Dimension

Indicators in the pressure dimension mainly consider urban natural pressure, which, in this paper, refers to floods. C1 and C2 are two indicators used to reflect susceptibility to urban flood disasters [49,50]. C3 reflects the hazard of the causative factors of urban flood disasters [51]. C4 reflects the stability of a disaster-prone environment during urban flood disasters [52].

### 2.3.2. Indicators in the State Dimension

Indicators in the state dimension primarily assess the quality and quantity of natural, economic, and social resources, including the state of regional and individual economic conditions, urban construction, and ecological environment, as well as demographic characteristics. In the social state, C5 and C6 reflect the vulnerability of the population and the adaptability of residents to flood disasters [29,53,54]. C7 reflects the number of people affected by flood disasters per unit area of urban land and the adaptability of residents to flood disasters [55]. In the economic state, C8 reflects the robustness and redundancy of the urban economy [56]. C9 and C10 reflect the robustness and redundancy of the individual economic resilience of urban residents [8,57]. As for the natural state, common indicators related to infrastructure, such as drainage systems, green coverage, and road systems are frequently mentioned to evaluate urban flood resilience [58–60]. C11 reflects the robustness of the spatial layout of urban drainage [61]; C12 reflects the ability of urban green spaces to purify surface runoff, promote the natural infiltration of rainwater, and store floodwater [62]; and C13 reflects the efficiency of the transportation system in response to urban flood disasters [59,63].

### 2.3.3. Indicators in the Response Dimension

Indicators in the response dimension take into account how people react to flood-related changes, including social and economic responses that reflect social learning and recovery abilities, as well as natural responses that reflect urban development and ecological recovery abilities. As for the social response, C14 and C15 reflect the government's strategic response to urban flood disasters [64–66]. C16 represents the city's capacity for learning and innovation, as well as its ability to reflect on flood disasters [18], and C17 reflects the ability of the public to make independent judgments, perform rescues, and recover from flood disasters [67,68]. As for the economic response, C18 reflects the resilience and redundancy of urban infrastructure in disaster recovery [69]. C19 reflects investment in disaster prevention and the social response capability of the city [70], and C20 reflects the adaptive capacity and flexibility of the urban economy [71]. As for the natural response, the indicators (C21 to C24) are mainly related to emergency resources, including water, transportation, power, and medical aid [72–74].

### 2.4. Hybrid Weighting Method

Because the evaluation indicators of urban flood resilience contain both objective data and subjective data, a combination weighting method that combines both subjective and objective methods is more appropriate for weighting calculations. For objective weighting,

the entropy method is a method that determines the weight of each indicator entirely based on the amount of information contained in each measurement indicator. It can make up for the utility difference caused by subjective weighting and fully reflect the information provided by the indicators of urban flood resilience. For subjective weighting, the analytic hierarchy process (AHP) mainly considers the importance of comparison between upper- and lower-level indicators but ignores the cross-relationships between indicators. Therefore, it needs to be combined with other methods. Since there may be dependency relationships between the various indicators of urban flood resilience, the superposition of these relationships may have an impact on the final result. The analytic network process (ANP) can consider the interrelationships between indicators. ANP is a decision-making method that is suitable for non-independent hierarchical structures proposed by Professor T.L. Saaty of the United States in 1996 [75]. It is a new practical decision-making method developed on the basis of ANP. It adopts a relative scale form and can make pairwise comparisons of the relative importance of elements at the same level and also measure the decision-making objectives according to the hierarchy, from top to bottom.

In summary, because of the inclusion of both objective data and subjective data in the measurement indicators of urban flood resilience, a combined subjective–objective approach, namely, the combination weighting method, is more suitable for calculating the weights. Because there may be dependence relationships between various urban flood resilience indicators, the combination of ANP and the entropy method can consider the interrelationships between indicators and levels, providing a new approach for weight calculation and helping to improve the objectivity and accuracy of measurement results. Therefore, combining ANP with the entropy method can effectively address the weaknesses of other methods, determine the weights of urban flood resilience indicators, and clarify the interrelationships between various indicators.

Thus, the general formula for calculating the combined weights of urban flood resilience can be provided as follows:

$$w_j = \frac{w_j^O w_j^S}{\sum_{j=1}^{24} w_j^O w_j^S}, \ \ j = 1, \, 2, \ldots, \, 24 \tag{1}$$

where $w_j$ represents the combined weight of the indicators, $w^O$ represents the objective weight obtained by the entropy method, $w^S$ represents the subjective weight obtained by the ANP method, and $j$ represents the number of indicators, which, in this paper, refers to the 24 indicators in the measurement framework of urban flood resilience in the Yangtze River Delta region.

*2.5. Evaluation Model Based on VIKOR*

The evaluation of flood resilience for the multiple cities in this paper is a multi-criteria decision-making (MCDM) problem, and therefore, the use of MCDM-related methods to construct a measurement model is the most appropriate. VlseKriterijumska Optimizacija I Kompromisno Resenje (VIKOR), which is a multi-criteria optimization and compromise solution method, is one of the MCDM technologies developed for the multi-criteria optimization of complex systems with initial weights. Similar to the technique of order preference similarity to the ideal solution (TOPSIS), VIKOR is also a compromise-sorting method based on the ideal point, considering the distance between the solution and the ideal solution. The optimal solution should be closer to the ideal solution and farther from the negative ideal solution. However, TOPSIS fails to consider the relative importance of positive and negative ideal solutions in decision-making, and VIKOR makes up for this deficiency by simultaneously considering the maximization of group utility and the minimization of individual regret in the decision-making process, making the decision-making more reasonable. Therefore, this paper chooses the VIKOR method to measure urban flood resilience.

In the VIKOR model, Ai (*i* = 1, 2, . . . , m) represents the cities being evaluated, where m = 27 in this case. Bj (*j* = 1, 2, . . . , n) represents the 24 resilience indicators of flood

resilience, where $n = 24$. *Xij* represents the value of each indicator for each city. The specific calculation steps are as follows:

(1) Standardize the indicators.

Normalize the original data results in a standardized matrix, V (Formula (2)), where $v_{ij}$ represents the standardized value of the $i$-th measured city on the $j$-th resilience indicator, calculated according to Formulas (3) and (4).

$$V = (v_{ij})_{27 \times 25} \tag{2}$$

For positive indicators, such as N8–N9, N11–N13, and N15–N25,

$$v_{ij} = \frac{x_{ij} - \min_i \{x_{ij}\}}{\max_i \{x_{ij}\} - \min_i \{x_{ij}\}} \tag{3}$$

For negative indicators, such as N1–N7, N10, and N14,

$$v_{ij} = \frac{\max_i \{x_{ij}\} - x_{ij}}{\max_i \{x_{ij}\} - \min_i \{x_{ij}\}} \tag{4}$$

(2) Determine the positive ideal solution ($X^+$) and negative ideal solution ($X^-$) for each indicator corresponding to each sample city.

$$X_i^+ = \max_j X_{ij} \tag{5}$$

$$X_i^- = \max_j X_{ij} \tag{6}$$

(3) Determine the group utility valve, $S_j$, and individual regret value, $R_j$, for each city, where $w_j$ represents the weight of the indicator.

$$S_j = \sum_{i=1}^n w_i \frac{(X_i^+ - X_{ij})}{X_i^+ - X_i^-} \tag{7}$$

$$R_j = \max_i w_i \frac{(X_i^+ - X_{ij})}{X_i^+ - X_i^-} \tag{8}$$

(4) Determine the compromise value, $Q_j$, for each city, where $S^- = min_j S_j$, $S^+ = max_j S_j$, $R^- = min_j R_j$, $R^+ = max_j R_j$; $v$ represents the adjustment coefficient between the group utility value and the individual regret value. When $v$ is greater than 0.5, it indicates a greater focus on the group utility value. When $v$ is less than 0.5, it indicates a greater focus on individual regret. Typically, $v$ is set to 0.5.

$$Q_j = v \frac{S_j - S^-}{S^+ - S^-} + (1 - v) \frac{R_j - R^-}{R^+ - R^-} \tag{9}$$

(5) Calculate the urban flood resilience (UR). The value of UR is between 0 and 1, and the larger the value, the greater the urban flood resilience.

$$UR = 1 - Q \tag{10}$$

(6) Justify the compromise value. Assuming that cities $A^{(1)}$ and $A^{(2)}$ are first- and second-rank cities, respectively, the compromise solution, $A^{(1)}$, should satisfy the following two conditions:

C1. Acceptable Advantage

$$U_R\left(A^{(2)}\right) - U_R\left(A^{(1)}\right) \geq 1/(n-1) \tag{11}$$

where $n$ represents the number of indicators.

C2. Acceptable Stability in Decision-Making

$S(A^{(1)})$ is the minimum value of individual regret, S, and $R(A^{(1)})$ is the minimum value of group utility, R.

(7) According to the above conditions, the final solution is determined as follows:

If all conditions are met, then $A^{(1)}$ is the final compromise value, which represents the city with the highest urban flood resilience.

If only condition C1 is met, then both $A^{(1)}$ and $A^{(2)}$ are the final compromise values, representing the city with the highest flood resilience.

If only condition C2 is met, then the compromise solution set $A^{(1)}$, $A^{(2)}$, ..., $A^{(r)}$ is obtained, where $A^{(r)}$ is determined by $U_R\left(A^{(1)}\right) - U_R\left(A^{(r)}\right) < 1/(n-1)$ with the maximum value of $r$.

### 2.6. Measurement Standard of UFR

Urban flood resilience can be classified into five levels, as shown in Table 2, based on the calculated values of Formula (10) after the compromise solution is verified.

**Table 2.** Measurement standard of urban flood resilience.

| Level of Urban Flood Resilience | Value Range of Urban Flood Resilience |
|---|---|
| Very High Resilience (VH) | $(0.80 < U_R \leq 1.00)$ |
| High Resilience (H) | $(0.60 < U_R \leq 0.80)$ |
| Moderate Resilience (M) | $(0.40 < U_R \leq 0.60)$ |
| Low Resilience(L) | $(0.20 < U_R \leq 0.40)$ |
| Very Low Resilience (VL) | $(0.00 < U_R \leq 0.20)$ |

With the measurement standard of the UFR and evaluation results from VIKOR, the ranks of each city can be obtained according to their specific level of UFR. In order the quantify the fluctuation of the UFR, the ranks of cities can be compared. By comparing the changes in the rankings of the UFR index between the beginning (2015) and the end (2019), all cities can be divided into three categories: relatively stable cities (RS, with rankings fluctuating between 0 and 3), cities with moderate fluctuations (MF, rankings fluctuating between 4 and 10), and cities with significant fluctuations (SF, rankings fluctuating at values over 10).

Furthermore, in order to analyze the changes in UFR over a continuous period of time, the annual change rates of the resilience indices of each city in the YRD region from 2015 to 2019 are also calculated. Combined with the overall change rate for the period of 2015–2019, the types of fluctuations in resilience were divided into four categories: (1) the gradually increasing type, mainly referring to the resilience index of a city gradually increasing over the study period; (2) the fluctuating increasing type, mainly referring to an overall upward trend in resilience over the study period, but with some downward trends in certain periods; (3) the fluctuating decreasing type, mainly referring to an overall downward trend in resilience over the study period, but with some upward trends in certain periods; (4) the gradually decreasing type, mainly referring to the resilience index of a city gradually decreasing over the study period.

### 2.7. Data Sources

The indicator data of the city's flood resilience measurement system come mainly from the following types of information. The statistical description of the indicators can be found in Table 3.

**Table 3.** Statistical description of the flood resilience indicators of the YRD.

| Indicators | Unit | Max | Min | Average | Standard Deviation |
|---|---|---|---|---|---|
| C1 | mm | 2619.80 | 405.70 | 1521.90 | 351.43 |
| C2 | mm | 328.70 | 34.80 | 112.76 | 49.71 |
| C3 | % | 0.08 | 0.00 | 0.03 | 0.02 |
| C4 | - | 0.96 | 0.09 | 0.53 | 0.28 |
| C5 | % | 35.00 | 15.09 | 21.75 | 3.76 |
| C6 | % | 19.73 | 9.86 | 13.82 | 2.56 |
| C7 | people/m$^2$ | 3826.00 | 192.42 | 821.35 | 659.04 |
| C8 | RMB | 174,270.00 | 28,808.00 | 88,845.80 | 35,174.73 |
| C9 | % | 55.30 | 16.85 | 37.71 | 4.88 |
| C10 | % | 3.93 | 0.00 | 0.65 | 0.53 |
| C11 | km/km$^2$ | 57.40 | 6.90 | 17.62 | 6.29 |
| C12 | % | 51.01 | 37.25 | 42.31 | 2.57 |
| C13 | m$^2$/people | 22.82 | 1.44 | 7.56 | 4.44 |
| C14 | - | 4 | 2 | 3.36 | 0.50 |
| C15 | - | 5.00 | 2.00 | 2.82 | 0.68 |
| C16 | % | 34.05 | 12.15 | 21.17 | 3.60 |
| C17 | % | 23.34 | 8.85 | 14.02 | 3.29 |
| C18 | RMB 10,000 | 8,635,958.00 | 26,781.00 | 1,047,939.28 | 1,503,934.45 |
| C19 | % | 117.38 | 7.41 | 43.24 | 29.33 |
| C20 | % | 69.90 | 26.98 | 48.25 | 9.01 |
| C21 | km/km$^2$ | 0.58 | 0.12 | 0.44 | 0.09 |
| C22 | car/10,000 people | 20.36 | 0.95 | 6.43 | 4.59 |
| C23 | % | 339.47 | 57.23 | 156.09 | 68.75 |
| C24 | bed/10,000 people | 112.99 | 13.92 | 52.92 | 17.79 |

(1) City yearbooks and bulletins: Statistical indicators (C5–C14 and C19–C23) in the indicator system are derived from the "China City Statistical Yearbook", the "Jiangsu Statistical Yearbook", the "Zhejiang Statistical Yearbook", the "Shanghai Statistical Yearbook", and statistical bulletins from various provinces and cities from 2015 to 2020.

(2) Meteorological data websites: The rainfall-related indicators (C1, C2, and C3) in the indicator system are derived from Chinese and provincial meteorological data websites, such as the Nanjing Meteorological Bureau website.

(3) Geographic data websites: The topographic and geomorphic feature indicators (C4) and water resource regulation capacity indicators (C22) in the indicator system are derived from various open-source geographic data websites, such as the Geographic Spatial Data Cloud website and Google Maps.

(4) Government-related websites: The evaluation indicators (C15 and C16) in the indicator system are derived from official government websites of various provinces and cities, such as the Shanghai Municipal People's Government website (http://www.shanghai.gov.cn/), the Nanjing Municipal People's Government website (http://www.nanjing.gov.cn/), the Jiangsu Provincial Bureau of Statistics website (http://tj.jiangsu.gov.cn/index.html), etc.

## 3. Results

### 3.1. Weighting Results

The weight calculation results of the ANP method show that the weight of resilience in the response dimension (0.4934) is the highest, followed by the state dimension (0.3108), and the lowest is the pressure dimension (0.1958). Among the pressure dimension indicators, flood risk (C3) has the highest weight; among the state dimension indicators, the drainage network condition (C11) has the highest weight; and for the response dimension, water resource regulation capacity (C21) has the highest weight. Among all indicators, flood risk (C3), water resource regulation capacity (C21), and emergency management capacity (C15) are the three indicators with the highest weights. As for the results of the entropy method, public transportation service capacity (C22), aging degree (C5), and social security capacity (C19) are the three indicators with the highest weights. By combining the objective and

subjective weighting results, the final weight results of urban flood resilience are shown in Table 4. Emergency management capacity (C15), urban maintenance and construction capacity (C18), and social security capacity (C19) are the three indicators with the highest weights.

**Table 4.** Weight of urban flood resilience indicators.

| Indicators | Objective Weight $w^O$ | Subjective Weight $w^S$ | Hybrid Weight $w_j$ | Ranks | Weight in the Dimension | Ranks in the Dimension |
|---|---|---|---|---|---|---|
| C1 | 0.017 | 0.020 | 0.010 | 21 | 0.073 | 4 |
| C2 | 0.032 | 0.053 | 0.049 | 9 | 0.364 | 1 |
| C3 | 0.012 | 0.128 | 0.044 | 11 | 0.329 | 2 |
| C4 | 0.048 | 0.023 | 0.032 | 16 | 0.236 | 3 |
| C5 | 0.076 | 0.037 | 0.081 | 2 | 0.246 | 1 |
| C6 | 0.034 | 0.013 | 0.013 | 19 | 0.038 | 6 |
| C7 | 0.028 | 0.043 | 0.035 | 14 | 0.106 | 4 |
| C8 | 0.042 | 0.049 | 0.060 | 8 | 0.183 | 3 |
| C9 | 0.012 | 0.018 | 0.006 | 24 | 0.020 | 9 |
| C10 | 0.053 | 0.005 | 0.007 | 23 | 0.021 | 8 |
| C11 | 0.034 | 0.080 | 0.079 | 3 | 0.242 | 2 |
| C12 | 0.025 | 0.048 | 0.035 | 15 | 0.105 | 5 |
| C13 | 0.061 | 0.007 | 0.012 | 20 | 0.037 | 7 |
| C14 | 0.026 | 0.026 | 0.020 | 17 | 0.037 | 10 |
| C15 | 0.023 | 0.102 | 0.068 | 6 | 0.127 | 4 |
| C16 | 0.021 | 0.059 | 0.036 | 13 | 0.067 | 9 |
| C17 | 0.057 | 0.011 | 0.018 | 18 | 0.033 | 11 |
| C18 | 0.067 | 0.034 | 0.065 | 7 | 0.122 | 5 |
| C19 | 0.072 | 0.044 | 0.091 | 1 | 0.170 | 1 |
| C20 | 0.035 | 0.008 | 0.009 | 22 | 0.016 | 6 |
| C21 | 0.024 | 0.111 | 0.078 | 4 | 0.144 | 2 |
| C22 | 0.077 | 0.016 | 0.036 | 12 | 0.068 | 8 |
| C23 | 0.059 | 0.042 | 0.072 | 5 | 0.133 | 3 |
| C24 | 0.065 | 0.024 | 0.046 | 10 | 0.085 | 7 |

*3.2. Results of the UFR Evaluation*

3.2.1. General Results

According to the classification of the UFR measurement levels in Section 2.6, the flood resilience index of each city in the YRD region from 2015 to 2019 can be classified into five categories. After classification, a yearly classification table of the flood resilience level of the cities in the YRD region can be obtained, as shown in Table 5. From 2015 to 2019, the flood resilience level of cities in the YRD region significantly improved. Among them, 23 cities had moderate resilience or above in 2019, accounting for 85.19%, with only 4 cities having low resilience, and the number of cities with relatively low resilience was 0. This means that, except for a few cities, the flood resilience levels of most cities in the research area have developed to a moderate level or above in the past five years, indicating that the flood resilience of cities in the research area is steadily improving.

The detailed results of the UFR in YRD during the calculation period are shown in Figure 4. From a regional perspective, the UFR index in the YRD region has shown a significant upward trend in the past five years. According to calculations, the average UFR index in the region was 0.4569 in 2015, but it dropped to 0.4243 in 2016, a decrease of about 7.14%. This may be due to heavy rainfall in 2016 [76]. During the summer of 2016, the entire YRD region was affected by continuous heavy rainfall, causing the mainstream of the Yangtze River, Dongting Lake, and Poyang Lake to exceed the warning water level, and many cities in Jiangsu, Anhui, and Zhejiang suffered from severe floods, resulting in huge losses. Therefore, the UFR index in 2016 showed a significant decrease. After that, the average UFR index values in the region from 2016 to 2019 were 0.4243, 0.4294, 0.4587, and 0.5539, showing an upward trend year by year. The increases in the UFR index were 1.21%, 6.81%, and 20.77%, respectively, indicating that the UFR in the YRD region is continuously

improving, and the magnitude of improvement is increasing, but the overall level is still at a moderate level.

**Table 5.** Classification of urban flood resilience levels of the cities in YRD.

| Measurement Standard | 2015 | | 2016 | | 2017 | | 2018 | | 2019 | |
|---|---|---|---|---|---|---|---|---|---|---|
| | Numbers | % | Numbers | % | Numbers | % | Numbers | % | Numbers | % |
| Very High Resilience | 0 | 0 | 0 | 0 | 0 | 0 | 0 | 0 | 3 | 11.11% |
| High Resilience | 4 | 14.81% | 2 | 7.41% | 1 | 3.70% | 5 | 18.52% | 7 | 25.93% |
| Moderate Resilience | 12 | 44.44% | 14 | 51.85% | 13 | 48.15% | 9 | 33.33% | 13 | 48.15% |
| Low Resilience | 11 | 40.74% | 10 | 37.04% | 12 | 44.44% | 13 | 48.15% | 4 | 14.82% |
| Very Low Resilience | 0 | 0 | 1 | 3.70% | 1 | 3.70% | 0 | 0 | 0 | 0 |

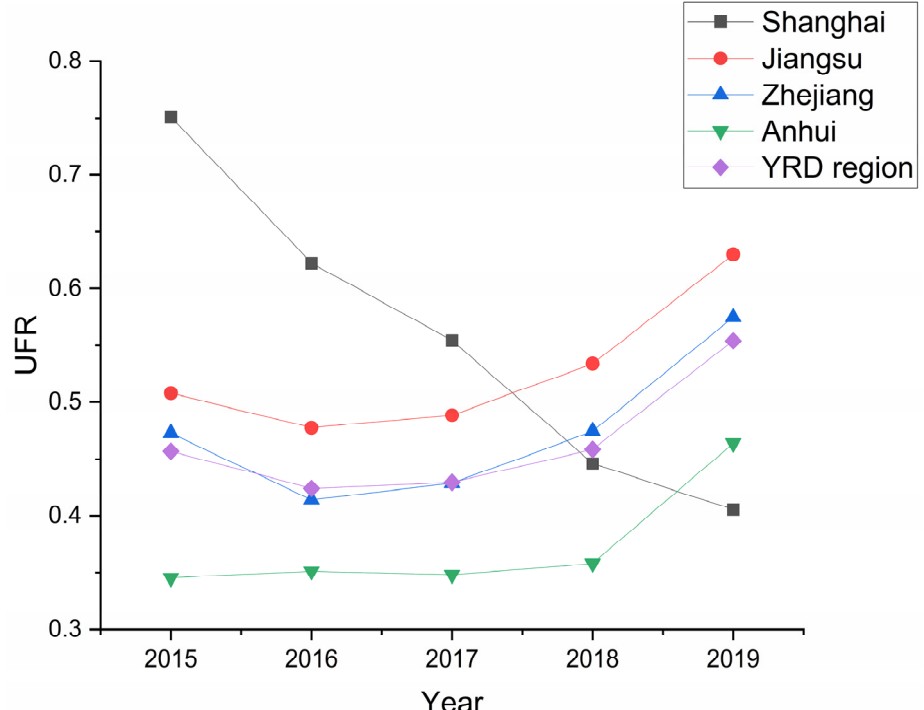

**Figure 4.** Urban flood resilience of provinces and the YRD region.

From a provincial perspective, Jiangsu, Zhejiang, and Anhui have shown the same development trend as the entire region, and their average urban flood resilience index has been rising year by year since 2016. In 2019, Jiangsu Province reached an average flood resilience index of 0.6299, rising from a moderate resilience level to a relatively high resilience level. Zhejiang and Anhui provinces still maintain a moderate resilience level, but the increase in the flood resilience index is higher than that of Jiangsu province. In contrast, Shanghai has shown a completely opposite trend in resilience development. Since 2015, the urban flood resilience index has declined year by year, with values of 0.7508, 0.6225, 0.5543, 0.4462, and 0.4056. The largest decline occurred between 2017 and 2018, reaching 19.5%. This trend is related to the rapid development of Shanghai: a large number of people have migrated to the city [77], and construction land continuously occupies green space, leading to a continuous decline in the urban flood resilience index.

From a municipal perspective, most cities have shown an upward trend in flood resilience over the years, as shown in Figure 5. Suzhou, Wuxi, and Changzhou, which are close in geography and have similar development situations, have shown a more consistent upward trend and reached a relatively high level of flood resilience in 2019. Among them, Suzhou has the most obvious upward trend and is at the relatively highest level among all the cities. Some cities, such as Nanjing, Nantong, Hangzhou, Ningbo, Huzhou, Shaoxing,

Jinhua, and Hefei, have gradually improved from a moderate resilience level to a high resilience level over the past five years. Except for Shanghai and Yancheng, the flood resilience index of other cities has increased to varying degrees. The fluctuation curves of urban flood resilience over the past five years show that there are differences in the level of urban flood resilience between cities in the Yangtze River Delta region, and the magnitude of fluctuations is also different.

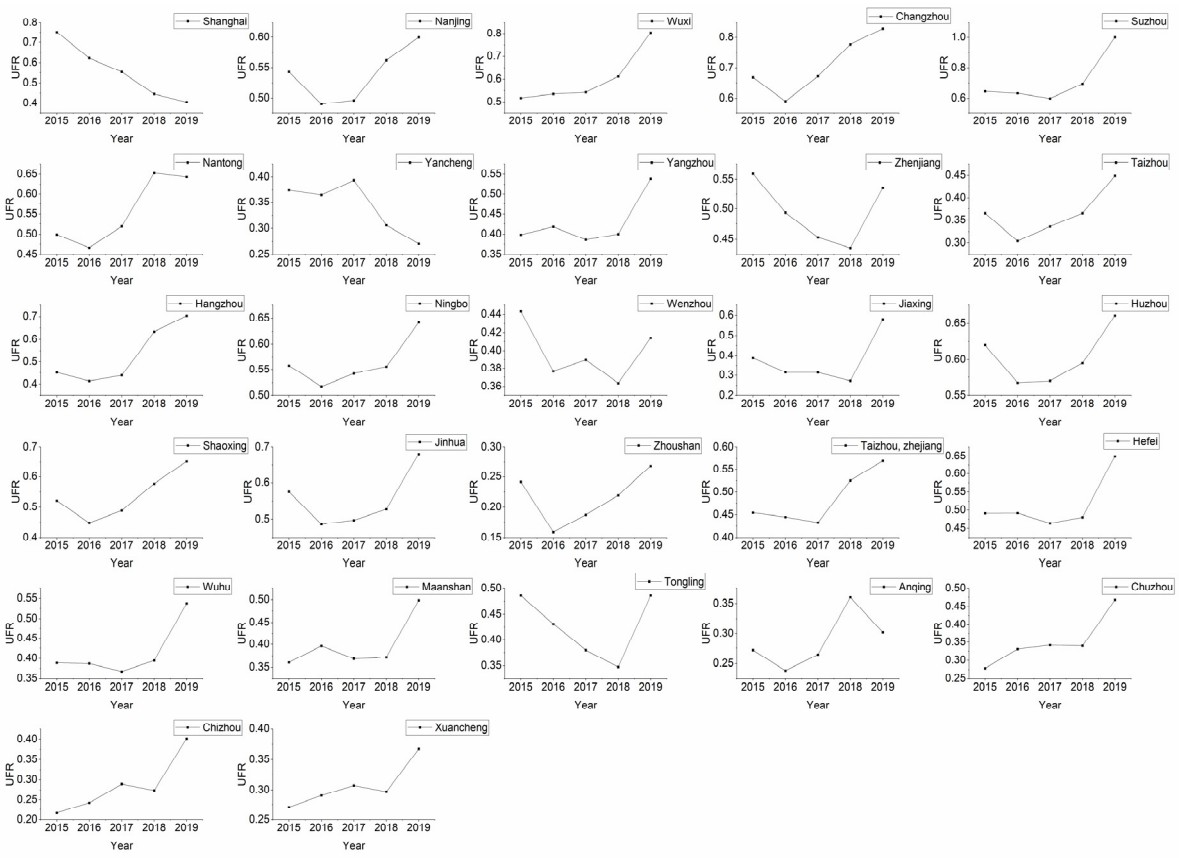

**Figure 5.** Urban flood resilience for cities in the YRD region.

### 3.2.2. Trends and Fluctuation Results

Based on the fluctuation classification defined in Section 2.6, it is found that, between 2015 and 2019, there were 15 relatively stable cities in terms of flood resilience, 10 cities with moderate fluctuations, and 2 cities with significant fluctuations as shown in Table 6. At the provincial level, cities in Anhui province showed more stable fluctuations in flood resilience compared with those in Jiangsu, Zhejiang, and Shanghai. At the city level, the rankings of 17 cities showed positive development (making up 62.96% of the total), while the remaining 10 cities showed varying degrees of decline in flood resilience rankings. Among them, Shanghai and Hangzhou had the most significant fluctuations, with Shanghai falling 21 places in 5 years, while Hangzhou rose 11 places. This result is related to the economic, social, and natural development characteristics of the two cities' urban systems.

**Table 6.** Changes in ranks of urban flood resilience indexes in Yangtze River Delta cities.

| YRD Cities | Rank in 2015 | Rank in 2016 | Rank in 2017 | Rank in 2018 | Rank in 2019 | Changes of Rank |
|------------|-------------|-------------|-------------|-------------|-------------|-----------------|
| Shanghai | 1 | 2 | 4 | 13 | 22 | −21 (SF) |
| Nanjing | 8 | 9 | 9 | 8 | 11 | −3 (RS) |
| Wuxi | 10 | 5 | 5 | 5 | 3 | 7 (MF) |
| Changzhou | 2 | 3 | 1 | 1 | 2 | 0 (RS) |
| Suzhou | 3 | 1 | 2 | 2 | 1 | 2 (RS) |

**Table 6.** *Cont.*

| YRD Cities | Rank in 2015 | Rank in 2016 | Rank in 2017 | Rank in 2018 | Rank in 2019 | Changes of Rank |
|---|---|---|---|---|---|---|
| Nantong | 11 | 11 | 7 | 3 | 9 | 2 (RS) |
| Yancheng | 20 | 20 | 15 | 23 | 26 | −6 (MF) |
| Yangzhou | 17 | 15 | 17 | 15 | 14 | 3 (RS) |
| Zhenjiang | 6 | 7 | 12 | 14 | 16 | −10 (MF) |
| Taizhou, JS | 21 | 23 | 22 | 18 | 20 | 1 (RS) |
| Hangzhou | 15 | 16 | 13 | 4 | 4 | 11 (SF) |
| Ningbo | 7 | 6 | 6 | 9 | 10 | −3 (RS) |
| Wenzhou | 16 | 19 | 16 | 19 | 21 | −5 (MF) |
| Jiaxing | 19 | 22 | 23 | 26 | 12 | 7 (MF) |
| Chaozhou | 4 | 4 | 3 | 6 | 6 | −2 (RS) |
| Shaoxing | 9 | 12 | 10 | 7 | 7 | 2 (RS) |
| Jinhua | 5 | 10 | 8 | 10 | 5 | 0 (RS) |
| Zhoushan | 26 | 27 | 27 | 27 | 27 | −1 (RS) |
| Taizhou, ZJ | 14 | 13 | 14 | 11 | 13 | 1 (RS) |
| Hefei | 12 | 8 | 11 | 12 | 8 | 4 (MF) |
| Wuhu | 18 | 18 | 20 | 16 | 15 | 3 (RS) |
| Ma'anshan | 22 | 17 | 19 | 17 | 17 | 5 (MF) |
| Tongling | 13 | 14 | 18 | 21 | 18 | −5 (MF) |
| Anqing | 24 | 26 | 26 | 20 | 25 | −1 (RS) |
| Chuzhou | 23 | 21 | 21 | 22 | 19 | 4 (MF) |
| Chizhou | 27 | 25 | 25 | 25 | 23 | 4 (MF) |
| Xuancheng | 25 | 24 | 24 | 24 | 24 | 1 (RS) |

Based on the change rate classifications, a chart depicting the changes in urban resilience levels in the YRD from 2015 to 2019 was created, as shown in Figure 6. The resilience levels of cities in the region have all undergone varying degrees of change between 2015 and 2019. Among them, there was only one city with a gradually increasing type, Wuxi, which benefited from relatively low flood risk, as well as the development of the economy and green infrastructure, resulting in a gradual increase in resilience over the years. There was also only one city with a gradually decreasing type, Shanghai, which was mainly due to the rapid urbanization and population growth, resulting in a decrease in ecological carrying capacity and weakened resistance to floods, leading to a significant decline in urban resilience. In total, 22 cities had a fluctuating increasing type, accounting for 81.48%, indicating that most cities have balanced the development of urbanization and flood disaster responses in recent years, and the resistance and disaster response capabilities of cities have continuously improved, resulting in an overall upward trend in urban resilience. The remaining three cities had a fluctuating decreasing type, accounting for 11.11%, i.e., Yancheng, Zhenjiang, and Wenzhou. These cities were closely connected to the fluctuating increasing type, as they were in a stage of rapid development, but their development and response to flood disasters were still in a process of adjustment, resulting in a significant fluctuation and decline in urban resilience.

As for each dimension, the changing trends can also be grouped into four categories. As shown in Figure 7, the evolution trend of UFR is influenced by changes in various dimensions. The gradually increasing type is primarily influenced by the rise of the resilience in the state and response dimensions; the fluctuating increasing type is mainly affected by the fluctuation and rise in pressure and response dimensions, as well as the ups and downs in the state dimension; the fluctuating decreasing type is mainly affected by the wavelike decline in the state dimension; and the gradually decreasing type is mainly affected by the gradual decrease in the state dimension. In terms of the pressure dimension, the number of cities with gradually increasing and fluctuating upward types is relatively high. All cities exhibit an upward trend in the response dimension, but the state dimension shows a polarization tendency, with nearly equal proportions of cities displaying upward and negative trends. Therefore, the improvement of flood resilience in most cities is due to the improvement of resilience in the pressure and response dimensions, while resilience in the state dimension remains the main obstacle to the improvement of UFR.

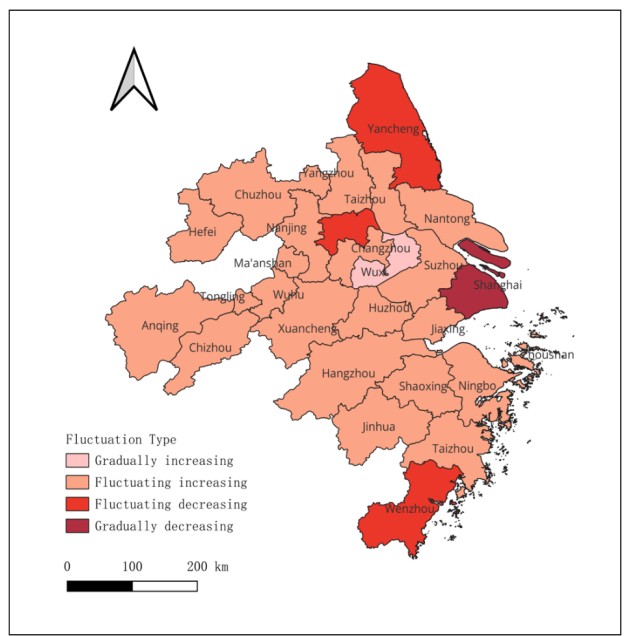

**Figure 6.** Fluctuation type of UFR for cities in YRD.

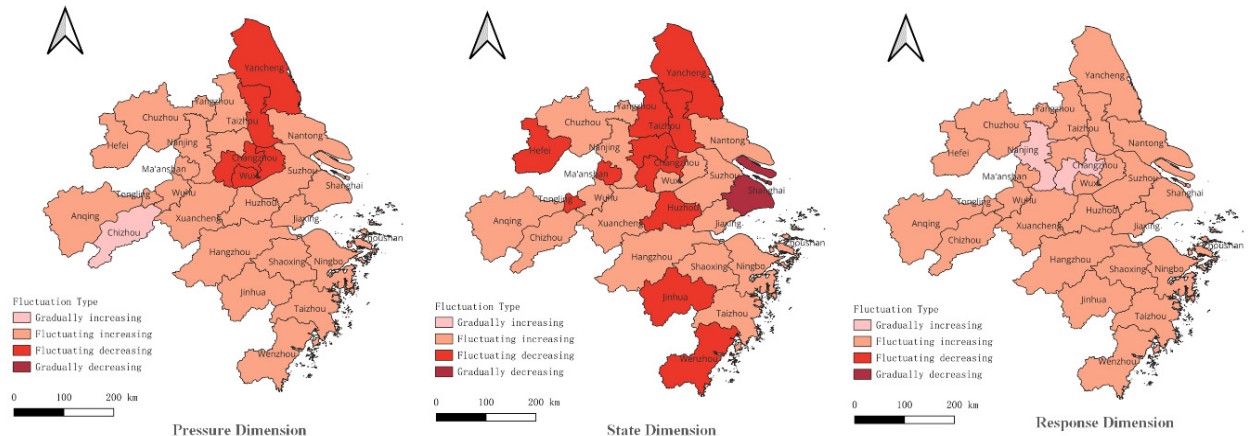

**Figure 7.** Three-dimensional fluctuation type of UFR in YRD cities..

## 4. Discussion and Implications

### 4.1. Sensitivity Analysis for UFR

As mentioned in Section 2.4, the adjustment coefficient, ν, between group utility and individual regret is generally set to 0.5, which means that there is no difference between group utility and individual regret and that the decision mechanism has reached a consensus. When ν > 0.5, the decision mechanism considers maximizing group utility (S); when ν < 0.5, the decision mechanism considers minimizing individual regret (R). If ν = 0, only R reflects the compromise value (Q), which reflects the urban flooding resilience index UR. If ν = 1, only S reflects the urban flood resilience index UR. Therefore, through sensitivity analysis and considering different ν values, different UFR results and their changes in the YRD region can be obtained. When ν = 0, individual regret (R) reflects the final compromise value (Q), which, in turn, reflects the level of UFR in each city. The larger the R value, the worse the corresponding indicator of the city's flood resilience. Therefore, by comparing the R values corresponding to ν = 0, the resilience evaluation indicators that need to be improved in each city can be identified, and then, the strategies for improving the resilience of each city can be guided.

As shown in Figure 8, the different colors in the graph correspond to different values of ν (considering a comparison interval of 0.1), and each ν value corresponds to a city's flood

resilience index UR. The results show that the urban flood resilience indexes of Suzhou, Jiaxing, and Hefei remain basically unchanged or changed very little (the change in the UR value with the ν value is within 0.05), indicating that these cities have the highest robustness. Among these cities, Suzhou has a relatively high level of urban flood resilience, Jiaxing has a medium level, and Hefei has a high level, indicating that the relative changes in their flood resilience are basically stable compared with other cities. In addition, Wuxi, Huzhou, and Shaoxing also exhibit good robustness (the change in the UR value with the ν value is within 0.1), and their urban flood resilience levels are not greatly affected by the ν value. The results show that the flood resilience levels of these cities have both the greatest group utility and the least individual regret, and the measurement results are relatively stable.

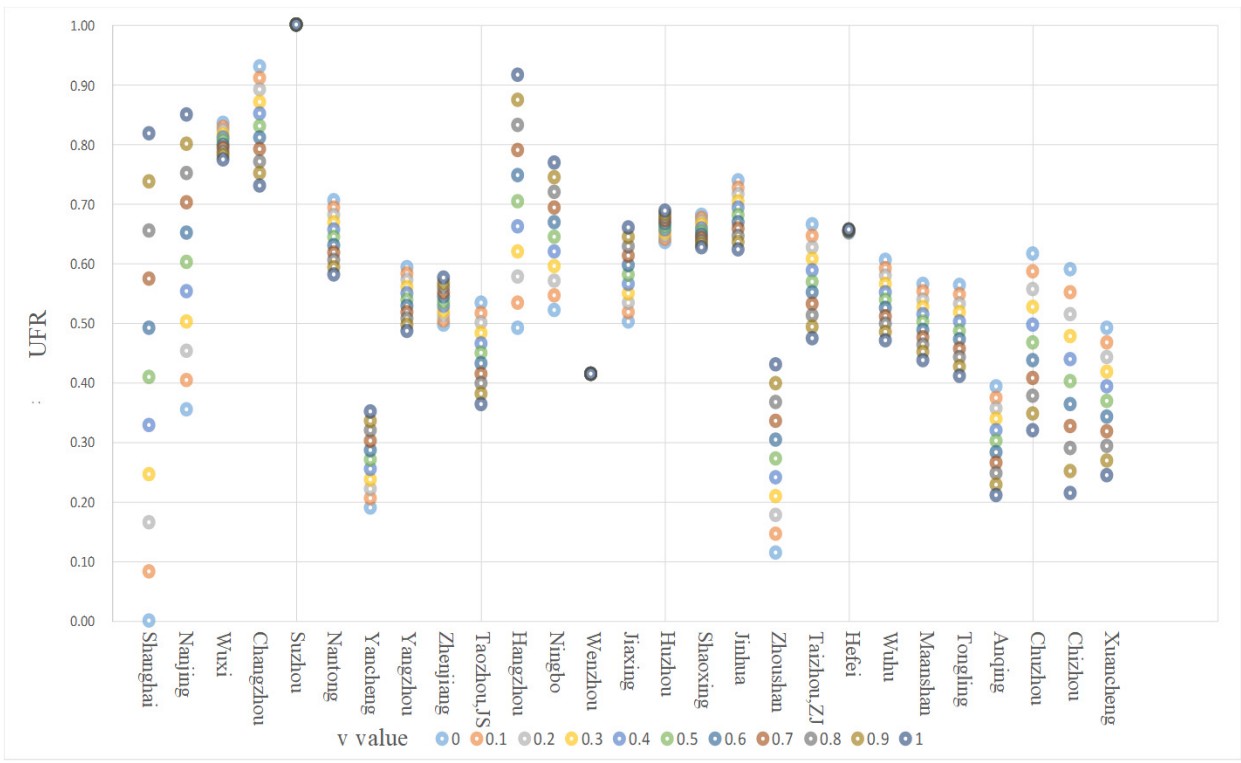

**Figure 8.** Sensitive analysis of UFR evaluations of Yangtze River Delta cities.

On the other hand, with the change in the ν value, the flood resilience index of other cities shows different trends, indicating that the results are affected by the group utility and individual regret of decision-makers. The urban flood resilience indexes of Shanghai, Nanjing, Yangzhou, Ningbo, Wenzhou, Huzhou, and Zhoushan increase with the increase in the ν value. This means that when decision-makers focus on maximizing group utility, the flood resilience index of these cities will increase; that is, in consideration of the overall YRD region, the flood resilience of these cities is relatively high. When the ν value decreases, the flood resilience indexes of other cities increase. Therefore, when decision-makers consider minimizing individual regret, these cities will show better urban flood resilience.

When the ν value is zero, the individual regret (R) reflects the compromise value (Q) and then reflects the flood resilience level of each city. Therefore, the city flood resilience index that needs to be improved can be clarified by comparing the R values corresponding to ν = 0. The largest individual regret value represents the worst value for each index, which urgently needs improvement in the corresponding city. For example, Shanghai performs the worst in the aging level (C5) and population exposure (C7) indicators. The aging level in Shanghai reached 35% in 2019, and the population density was 3823 people/square kilometer, both of which were the highest among 27 cities [78]. Improving these two aspects is essential for enhancing Shanghai's urban flood resilience.

Therefore, by analyzing the individual regret values of each city's corresponding indicators, the city flood resilience indicators that need to be improved urgently can be summarized as shown in Table 7. Each city can develop corresponding strategies and measures for improving urban flood resilience based on the indicators that need to be improved. Among them, some cities need to focus on improving certain indicators, such as Nanjing and Changzhou, which are lacking in flood warning capabilities, and Yangzhou, which needs to further improve its emergency management capabilities. Some cities need to improve multiple indicators, such as Anqing, which needs to improve its regional economy, urban roads, social security capabilities, communication capabilities, and medical assistance capabilities; vigorously develop the economy and infrastructure construction; and enhance its ability to resist and respond to flood disasters.

**Table 7.** Urban flood resilience indicators that urgently need to be improved by cities in YRD.

| Indicators That Need to Be Improved | Cities | Indicators That Need to Be Improved | Cities |
|---|---|---|---|
| C1: Climate change | Wenzhou | C13: Urban road conditions | Yancheng, Wenzhou, Jiaxing, Jinhua, Tongling, Anqing, Xuancheng |
| C2: Heavy short-term rainfall | Chizhou | C14: Flood disaster early warning ability | Nanjing, Changzhou |
| C3: Flood risk | Yancheng | C15: Emergency management ability | Yangzhou |
| C4: Topographic features | Chizhou | C16: Knowledge learning ability | Zhoushan |
| C5: Aging degree | Shanghai | C17: Public reaction ability | Taizhou, Xuancheng |
| C6: Children percentage | Wenzhou | C1: Urban maintenance and construction budget | Tongling, Chizhou |
| C7: Population exposure to floods | Shanghai | C19: Flood insurance | Anqing, Chizhou |
| C8: Regional economic status | Anqing | C20: Economic diversity | Chuzhou |
| C9: Residents' economic status | Ma'anshan | C21: Water and power supply | Zhoushan |
| C10: Employment status | Hefei | C22: Public transportation service capacity | Anqing, Chuzhou, Chizhou, Xuancheng |
| C11: Drainage network | Yancheng | C23: Communication capacity | Anqing |
| C12: Green coverage | Jiaxing | C24: Medical aid capacity | Anqing, Chuzhou |

*4.2. Implications for Flood Resilience Improvement*

4.2.1. Implications for the YRD Region

According to the UFR evaluation results, the overall UFR of the region is at a moderate level and shows an upward trend, but it is significantly affected by rainfall, and resilience in the pressure dimension fluctuates greatly, requiring long-term dynamic monitoring and calculation. The YRD region is densely covered with rivers, with a developed water system [38]. Under extreme rainfall scenarios, besides the Yangtze River, there is a possibility of flooding from tributary lakes and rivers such as Tai Lake, West Lake, Chao Lake, the Huangpu River, and the Qinhuai River [79]. In order to improve the UFR and promote the coordinated development of resilience in the pressure, state, and response dimensions, an overall regional flood map is suggested to be put forward. Developed cities in the world have all come up with their own flood maps, such as the New York Flood Map and the London Thames CFMP Plan [80,81]. These maps, including regional flood risk maps, flood risk diagnosis, and resilience enhancement paths, can be used as a guide [82], which helps to clarify existing and potential problems, deepen the overall understanding of regional flood disasters, guide future planning, and promote the implementation of relevant policies.

Furthermore, considering the current lack of resilience-related content in the policies of the YRD region [83], there is a need to increase the priority of climate change adaptation in urban governance. Although current regional planning has focused on environmental, education, medical, and technological issues, compared with controlling greenhouse gas emissions and building smart cities [34], climate change adaptation and resilience improvement have not received sufficient attention in the governance of the YRD. The region is

mainly a subtropical monsoon climate and faces severe climate change in the future [17]. Therefore, climate change adaptation should be included in regional development planning to improve the region's ability to respond to climate change, flood catastrophes, and other associated challenges through a top-down approach. We suggest strengthening the role of resilience in urban planning and governance and developing a detailed regional climate action plan to enhance UFR.

### 4.2.2. Implications for the Provinces

In addition to implementing the overall regional strategy discussed above, the "Three Provinces and One Municipality" in the YRD should develop overall improvement strategies tailored to the specific situations of their respective UFR level in accordance with the internal needs of each province for improving flood resilience in conjunction with current policies.

As for the municipality Shanghai, although it is the leading city in the economy of the YRD region, its UFR is relatively moderate within the region. The overall resilience in the state dimension shows a trend of decline, which should be the focus of its resilience strategy. Because Shanghai has the highest aging population level in the YRD and is the only megacity in the region, its population exposure level is relatively high and urgently needs to improve its social resilience. The "Shanghai City Master Plan (2017–2035)" also pointed out the need to strictly control the city's permanent population and control the population size. Compared with other provinces in the YRD, Shanghai is relatively less affected by floods, but the increasing population may lead to more problems, such as land, transportation, water use, energy consumption shortages, etc. [27]. Shanghai currently includes resilient city development in its overall work deployment and has proposed building a "more sustainable and resilient ecological city" [84,85]. Under the supervision of regional resilience strategies, this must play to its strengths while avoiding vulnerabilities in order to take a leading role in the YRD region.

As for the other three provinces, Jiangsu province is currently at a relatively high level of resilience. The flood resilience of the three cities in Suzhou, Wuxi, and Changzhou is at a relatively high level, exerting a positive radiating effect on surrounding cities. Therefore, Jiangsu province should, based on its current level of urban development, incorporate the resilience concept into its new "strong, rich, beautiful, and high" blueprint, guided by the aforementioned flood resilience strategy in the YRD region. To address flood disasters, provincial flood resilience policies and strategies should be developed while also steadily pushing urban resilience construction; balancing social, economic, and environmental growth; and building a resilient Jiangsu to deal with future climate change.

The UFR in Zhejiang province is at a relatively moderate level within the YRD region, but the development of resilience within the province is not well coordinated. There are significant differences in the pressure, status, and response dimensions between the southern and northern parts of Zhejiang, which echoes the rapid regional disparity within the province since the reform and opening up of China [86]. Additionally, according to the assessment of resilience in the pressure dimension and the data from water and drought disaster reports in previous years, the coastal areas of Zhejiang Province have also been greatly affected by floods and waterlogging. To close the gap between cities within the province, Zhejiang province is encouraged to engage in top-level design and incorporate flood resilience building into regional coordinated development based on the aforementioned regional strategy.

Compared with other provinces in the YRD region, the UFR of Anhui province is at a relatively low level, with both low resilience in the state and response dimensions. Cities such as Anqing, Chizhou, and Tongling have long been in the low cluster of flood resilience, and most cities in the province face the most severe flood risk. Furthermore, Anhui is also the largest flood discharge area in the YRD [40]. Therefore, improving urban flood resilience in Anhui province is of the highest urgency and priority, and it is necessary to comprehensively enhance flood resilience at the overall level and then address specific issues through targeted strategies. Possible strategies and measures can be

considered, including advancing the comprehensive flood management mode of human–water harmony, managing middle and small rivers, and strengthening key basic research on flood control and disaster reduction [87,88].

### 4.2.3. Implications for the Cities

Each city in the YRD region has unique natural and geographical environments and social histories, which require specific analysis for the improvement of UFR. Therefore, under the premise of following regional and provincial development requirements and strategies, the corresponding enhancement strategies should be tailored to each city's specific circumstances. Each city should carefully assess its unique development characteristics and current level of UFR, utilizing benchmarking to identify areas for improvement. Subsequently, cities should propose corresponding measures aligned with the policies and requirements of YRD integration, aiming to enhance their urban flood resilience.. Detailed suggestions can be found in Table 8.

**Table 8.** Improvement strategies for each city in the YRD.

| Cities | Priority Strategy | Cities | Priority Strategy | Cities | Priority Strategy |
|---|---|---|---|---|---|
| Shanghai | Control aging and population density growth, improve social resilience in the state dimension | Taizhou | Enhance disaster early warning capability; strengthen knowledge learning and emergency management abilities | Taizhou | Control population, improve emergency and early warning capabilities |
| Nanjing | Strengthen municipal infrastructure resilience | Hangzhou | Control aging, balance residents' employment and income | Hefei | Develop urban economy |
| Wuxi | Comprehensively improve natural, economic, and social resilience in the response dimension | Ningbo | Strengthen urban infrastructure construction | Wuhu | Improve emergency, early warning and learning capabilities |
| Changzhou | Improve infrastructure resilience | Wenzhou | Strengthen flood control and reduce flood risk | Ma'anshan | Develop urban economy and improve residents' income |
| Suzhou | Control aging and strengthen urban municipal infrastructural resilience | Jiaxing | Protect the ecological environment, pay attention to the disaster-prone environment, reconsider urban planning | Tongling | Consider the flood risk map in urban planning, develop the urban economy |
| Nantong | | Huzhou | Improve the disaster early warning and emergency response capabilities | Anqing | |
| Yancheng | Control air pollution and strengthen urban municipal infrastructural resilience | Shaoxing | Control the population density | Chuzhou | |
| Yangzhou | Enhance disaster early warning capability; strengthen knowledge learning and emergency management abilities | Jinhua | Reconsider the urban planning | Chizhou | |
| Zhenjiang | Control aging and population density growth, strengthen infrastructural resilience | Zhoushan | Improve the learning and recovery abilities related to disasters | Xuancheng | Improve infrastructure resilience and improve emergency, early warning, and learning abilities |

Furthermore, considering the national development plan, the high-resilience cities in the YRD region are encouraged to strengthen their radiating effect on their surrounding neighboring cities and drive the relevant metropolitan areas to improve their flood resilience. Cities with a moderate level of resilience should plan ahead of time and learn from the

experiences of advanced cities. Because moderate-resilience cities account for a relatively large proportion, these cities can be prioritized in regional planning. Low-resilience cities should begin with their current condition, do a good job of responding to flood disasters, make up for inadequacies, and then examine overall flood resilience improvement.

## 5. Conclusions

This paper proposed a conceptual framework for urban flood resilience evaluation, combining the PSR model and SENCE theory. The indicators measure natural, economic, and social resilience from the pressure, state, and response dimensions in the stages of the flood disaster cycle. The proposed evaluation framework was applied to 27 cities in the YRD region. The results showed a notable upward trend of UFR for the region throughout the period. Jiangsu, Zhejiang, and Anhui provinces have shown the same gradual development trend, while Shanghai showed a totally different trend. The cities were grouped into different categories according to their changing resilience trends. In total, 81.41% of cities showed a fluctuating increasing trend in urban flood resilience during the period, while only Shanghai showed a gradually decreasing trend.

The relative evaluation method VIKOR was employed in this paper to assess the balance among all the cities as well as within individual cities, making the results more rational and compelling. The sensitivity analysis revealed that Suzhou, Jiaxing, and Hefei have the highest robustness, as their UFR scores are almost unchanged or vary very little. When considering the individual regret value, it is also useful to show which indications in each city need to be improved immediately. Following the entire process, the regional, provincial, and city-level implications can be easily provided, along with suggested guidelines for further flood resilience improvement. Cities may establish more effective flood resilience plans and build sustainable and adaptive urban landscapes by using an integrated strategy that addresses social, economic, and natural aspects.

The proposed methodology and evaluation procedure are simple to implement and useful for measuring flood resilience standards; identifying flaws; and providing improvements at the regional, provincial, and city levels. They clearly demonstrate the shifting trends, strengths, and weaknesses of each city, allowing for the development of specific measurements for each city within the context of the entire ecosystem. They can also be applied to international regions or cities to investigate UFR trends, identify gaps, and recommend flood prevention and resilience enhancement methods. UFR under pandemic situations can be considered in future works.

**Author Contributions:** Conceptualization, S.Z. and H.F.; methodology, software, validation, S.Z.; formal analysis, Q.S.; investigation, resources, data curation, writing—original draft preparation, S.Z. and Q.S.; writing—review and editing, H.F.; visualization, S.Z.; supervision, H.F.; funding acquisition, S.Z. All authors have read and agreed to the published version of the manuscript.

**Funding:** This research was funded by the National Natural Science Foundation of China (Grant number: 72204127).

**Data Availability Statement:** Not applicable.

**Conflicts of Interest:** The authors declare no conflict of interest.

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
