# Peer review of "Evaluating Urban Flood Resilience within the Social-Economic-Natural Complex Ecosystem: A Case Study of Cities in the Yangtze River Delta"

_land, doi:10.3390/land12061200_

Round 1
Reviewer 1 Report
Excellent job.
I have only two minor observations:
1) I suggest putting a graticule showing coordinates in figure 1 (line 106) to help readers locating the study zone. It would be better to put a World Geographic coordinate system (latitude and longitude) as the common system use in the entire world. The other maps in the paper do not need a graticule if figure 1 already shows the coordinate system.
2) In line 165, the text shows that response dimension refers to people reaction. It should be clarified that "people" represents authorities in a a community (responsible of decision-making process) rather than a group of individuals. Is that true? or am I understanding wrong?
Author Response
Comment 1: I suggest putting a graticule showing coordinates in figure 1 (line 106) to help readers locating the study zone. It would be better to put a World Geographic coordinate system (latitude and longitude) as the common system use in the entire world. The other maps in the paper do not need a graticule if figure 1 already shows the coordinate system.
Response 1: Thank you so much for this suggestion. We have revised the map with a world geographic coordinate system. Please refer to Figure 1.
Comment 2: In line 165, the text shows that response dimension refers to people reaction. It should be clarified that "people" represents authorities in a community (responsible of decision-making process) rather than a group of individuals. Is that true? or am I understanding wrong?
Response 2: Thank you for this comment. Indicators in the response dimension include C14 to C17. Among them, C14 to C16 refer to the authorities in a community (responsible of decision-making process) and C17 refers to the individuals’ reaction. We added some explanations in this part, please refer to line 172-176.
Reviewer 2 Report
The article is interesting. It raises an important issue of estimating urban flood resilience. The work is described clearly, supported by rich and up-to-date literature. The structure of the article and the method of conducting the research is correct. I only have a few comments on the article, listed below:
1. There is no reference to Figure 3 in the text. It would be worth describing what is contained in this figure. I would also suggest moving it before point 2.3.1 of the work, for example, placing it under Table 1, where there is a mention of the number of indicators (the figure shows them all).
2. In Figure 5, the location of city names along the vertical axis is misleading. The names might look better above the charts.
3. It seems that in Figure 8 the city descriptions are shifted along the horizontal axis. Also, 26 cities are listed when there should be 27. Please check it out.
4. Line 125: "UFR" instead of "FUR".
Author Response
Comment 1: There is no reference to Figure 3 in the text. It would be worth describing what is contained in this figure. I would also suggest moving it before point 2.3.1 of the work, for example, placing it under Table 1, where there is a mention of the number of indicators (the figure shows them all).
Response 1: Thank you so much for this suggestion. We moved Figure 3 before the point 2.3.1, under Table 1. The references for Figure 3 are in the section 2.3.1 to 2.3.3, where we give references and explanations for each of the indicator.
Comment 2: In Figure 5, the location of city names along the vertical axis is misleading. The names might look better above the charts.
Response 2: Thank you so much for this suggestion. We have revised the Figure 5 according to your comment.
Comment 3: It seems that in Figure 8 the city descriptions are shifted along the horizontal axis. Also, 26 cities are listed when there should be 27. Please check it out.
Response 3: Thank you so much for this comment. We have revised Figure 8. Please refer to line 470-471.
Comment 4: Line 125: "UFR" instead of "FUR".
Response 4: Thank you so much for this comment. We have revised it. Please see Line 125.